# Validation of the somatic symptom disorder—B Criteria Scale (SSD-12) in Bangladesh

**Sumaiya Habib, Muhammad Kamruzzaman Mozumder** *

Department of Clinical Psychology, University of Dhaka, Bangladesh

* mozumder@du.ac.bd

## Abstract

### Background

The absence of a reliable and valid Bangla instrument for measuring somatic symptom disorder hinders research and clinical activities in Bangladesh. The present study aimed at translating and validating the Somatic Symptom Disorder-B criteria (SSD-12).

### Method

A cross-sectional design was used with purposively selected clinical (n = 100) and non-clinical (n = 100) samples. The clinical sample was collected from psychiatric departments at three hospitals, while the non-clinical sample was drawn from the local community. Exploratory (EFA) and confirmatory factor analysis (CFA) were conducted on the SSD-12, along with reliability and validity assessments.

### Results

Results indicated satisfactory internal consistency reliability (Cronbach's alpha = .94, split-half r = .93); criterion-related validity (r = .86, with Morey's Somatic Complaints Scale); and construct validity (r = .64 with anxiety subscale and r = .57 with depression subscale of the Hospital Anxiety and Depression Scale) of the translated scale. In contrast to the three-factor structure of the original SSD-12, the Bangla version indicated a single-factor structure (accounting for 61.29% of the total variance). This scale also demonstrates its ability to distinguish between clinical and non-clinical participants ($t_{198}$ = 16.74, p < .01).

### Conclusion

The Bangla translated SSD-12 has demonstrated strong psychometric properties, indicating its suitability for use in Bangladesh. This tool is expected to aid mental health practitioners in their clinical work by providing them with a quick assessment of their patients having somatic complaints.

**Data Availability Statement:** The complete dataset used in this publication are available at https://doi.org/10.17605/OSF.IO/RH5BK.

**Funding:** Collection of data was partially funded by Bangladesh National Science and Technology

(NST) Fellowship received by SH. However, the author(s) did not receive any funding for preparing or publishing the manuscript.

**Competing interests:** The authors have declared that no competing interests exist.

## Introduction

Patients at all stages of healthcare, frequently experience physical complaints such as pain in various body parts, exhaustion, or other problems with the bodily systems. The severity spectrum of these complaints and sufferings may range from minor signs with little functional impairments to diseases that are entirely incapacitating. However, a significant portion of these presentations are the result of a psychogenic process [1]. Psychiatric diagnoses such as Somatic Symptom Disorder (SSD) [DSM-5; 2] or Bodily Distress Disorder [ICD-11; 3] are often missed for these patients [1]. This misdiagnosis often contributes to lengthy and ultimately ineffective therapy resulting in a substantial increase in the cost of treatment, burden of disease, and distress in the treating physicians [4]. Early and accurate identification of the disorder is therefore crucial for somatic symptoms.

Somatization disorders are characterized by a lifetime history (generally beginning before age 30) of seeking treatment for or becoming impaired by multiple physical complaints that cannot be fully interpreted by medical causes and are not intentionally feigned [5]. Somatization is generally characterized by three distinct features [6]. Firstly, it presents with medically unexplained somatic symptoms such as pain, weakness, or shortness of breath. Secondly, patients exhibit somatic preoccupation or hypochondriacal worry demonstrating excessive engagement (in terms of time and effort), and concerns about their bodily symptoms. These concerns often are overtly disproportionate and may be accompanied by excessive body checking and reassurance-seeking [7]. Thirdly, patients may present these somatic symptoms as part of the clinical manifestations of anxiety, affective, or other psychiatric disorders [6].

Somatization may present with a range of symptoms, Bangladeshi children and adolescents diagnosed with somatoform disorder report an average of 12 to 16 somatic symptoms [8]. Pain is the most common symptom among patients. In Bangladesh, abdominal pain is the most frequent somatic symptom in children and adolescents [8], while a German study of 7,925 adults aged 40 to 80 years found that pain complaints (arms, legs, joints, back pain) were most common, followed by back pain, headaches, nausea, constipation/diarrhea, shortness of breath, dizziness, and heart racing or pounding [9].

Across the globe, including Bangladesh, somatic symptom disorders have been reported to be the third most prevalent mental disorder after anxiety disorders and depression [10, 11]. Mohit (12) found that 6.6% of the patients from the outpatient department at the specialized mental hospital suffer from somatoform disorder, while a recent nationwide survey revealed a prevalence of about 2.1% among the adult population (105 million) in Bangladesh [11].

Somatoform disorders are among the most common psychiatric disorders in general medical settings. Somatoform disorders often coexist with other comorbidities, with 8% of primary care patients meeting the criteria for 'multi-somatoform disorder'[13]. Depression, conversion disorder, hypochondriasis, somatization, and pain disorders are the most common comorbid conditions associated with somatization [14]. Apart from comorbidity, somatization disorder has been reported to be strongly associated with depression and anxiety, moderately with schizophrenia and mania, and weakly with substance use and antisocial personality [15]. Accurate diagnosis, support, and reassurance are the cornerstones of the treatment of somatization disorder [16]. Approaches typically involve psychotropic medications and psychological therapies (such as, cognitive behavior therapy) focusing on cognitive, emotional, and behavioral aspects [17]. The prognosis for somatic symptom disorder shows improvement in 50–75% of patients, while 10–30% experience a worsening of their condition under combination treatment (medication and psychological therapy) [18]. Fewer symptoms and higher baseline functioning have been linked with a better prognosis [19].

An accurate identification and assessment of somatic symptom disorders in a clinical setting is necessary for effective intervention. However, the assessment of SSD is challenging due to a variety of diagnostic difficulties including symptom overlap. Such overlap may be observed with medical conditions [20] and with other mental illnesses such as illness anxiety disorder, conversion disorder, psychological factors affecting medical conditions, and factitious disorder [21]. Numerous questionnaires such as the somatization subscale of the Symptom Checklist-90 [22] and the Patient Health Questionnaire (PHQ-15) [23] are available for assessing somatic symptoms in the general population. However, none of them are suitable for diagnostic purposes as they fail to capture psychological attributes.

In clinical settings, the Structured Clinical Interview for DSM-5 [SCID-5; 24] is considered the gold standard for diagnosing DSM-5 disorders. However, as a time-consuming process, conducting SCID is often not feasible, especially in resource-constrained settings such as Bangladesh. Although the recently available Bangla tool "Morey's Somatic Complaints Scale (SCS)" [25] is utilized for assessing somatic complaints as part of a larger Personality Assessment Inventory (PAI), it is not designed or suitable for diagnosing or assessing criterion B of somatic symptom disorder. The Somatic Symptom Disorder–B Criteria Scale [SSD-12; 20] is a brief self-report tool specifically designed to evaluate DSM-5 criterion B of the somatic symptom disorder and hence is suitable for diagnosing the disorder. The SSD-12 assesses patients' cognitive, affective, and behavioral symptoms and therefore, has additional clinical utility along with diagnostic use. Translation and validation of the SSD-12 scale in Bangla seemed prospective for its clinical as well as research application with SSD patients. It is expected that the Bangla version of the instrument will assist in prompt diagnoses leading to early and accurate intervention, which will enhance the quality of life of the patients.

## Methods

### Participants

This study employed a cross-sectional design. Purposive sampling was used to recruit 200 participants (100 clinical, 100 non-clinical). The clinical sample was collected from the psychiatric departments of three hospitals while the non-clinical sample was collected from the general population from the local community. The clinical participants, diagnosed by psychiatrists, met DSM-5 diagnostic criteria for somatic symptom disorder, while the non-clinical participants had no history of any psychiatric disorders. The inclusion criteria required the participants to be adults (i.e., age above 18 years) and to meet the group criteria (diagnosis of somatic symptom disorder for the clinical group and no mental illness for the non-clinical group). The exclusion criteria included the presence of active suicide risk (< 3 months), intoxication with substance, intellectual disability, and inability to communicate in Bangla. Demographics revealed the predominance of females (63%) over males (37%) in the total sample who had an age range of 18 to 65 years (M = 30.02, SD = 8.99). Among the participants, 17% completed primary education, 39% held bachelor's degrees, and 26% held master's degrees or equivalent. The majority (66.5%) of them described their economic condition as average, while 21% described it as below average. Data for this study were collected from the psychiatric departments of three tertiary hospitals, providing access to a diverse patient population. For one of the hospitals, the majority of the data were collected from the inpatient department, while for another, the majority came from the outpatient department. These differences may result in inadvertent variation in the patient groups across the three settings. However, as no comparisons between the settings were planned, we believe these variations rather contributed to the generalizability of the findings.

## Instruments

A custom-built socio-demographic information questionnaire was used to collect information about the participants' age, sex, education, marital status, and socio-economic status. Additionally, three established scales were used to measure different constructs as part of the validation process of the SSD-12. Participants were instructed to complete the SSD-12 questionnaire based on their experiences over the last one week, aiming to capture the severity of recent symptoms and their impact.

Somatic Symptom Disorder–B Criteria Scale [SSD-12; 20]. The SSD-12 consists of 12 items distributed over cognitive, affective, and behavioral domains that reflect in a three-factor structure [20]. The respondents report their experiences of cognition, emotion, or behavior on a 5-point Likert scale such as, "I think that my physical symptoms are signs of a serious illness", "I am very worried about my health", "My health concerns hinder me in everyday life". The total score ranged from 0 to 48 with higher scores indicating higher severity of SSD. Adequate reliability (Cronbach's alpha = .95) of the SSD-12 has been reported, along with a high correlation between somatic symptoms (r = .47, p < .01), depressive symptoms (r = .22, p < .01), and health anxiety (r = .71, p < .01), confirming the validity of the test [20].

Morey's Somatic Complaints Scale [SCS; 26]. The SCS has three subscales assessing conversion, somatization, and hypochondriasis [26]. The somatization subscale of the Bangla SCS [25] was used as the criterion measure in evaluating the construct validity of the SSD-12. Satisfactory psychometric properties where construct validity at r = .925, (with GHQ-28) and Cronbach alpha at .97 of the Bangla SCS have been reported [25].

Hospital Anxiety & Depression Scale [HADS; 27]. The HADS [27] is a widely recognized and commonly used self-report instrument that measures depression and anxiety as two distinct dimensions, each with 7 items, using a 4-point response scale ranging from 0 (no distress) to 3 (significant distress). This scale was found to perform well in assessing the severity of anxiety and depression in both somatic and psychiatric cases and the general population [28]. It was hypothesized that individuals with higher somatic complaints would have a higher level of anxiety and depression [29] and hence, the HADS was used to assess criterion validity of the Bangla SSD-12.

## Procedures

Translation and back-translation were done following the International Test Commission's guidelines [30] in preparing the Bangla version of the SSD-12 scale. Two bilingual experts translated the questionnaire from English to Bangla, and their translations were synthesized into one version, which was then back-translated into English by another bilingual expert. The back-translation was compared with the original English version by the original author of SSD-12, and no discrepancies in the meaning of any items were reported. Details on the processes used in the translation and validation of the SSD-12 are presented in Fig 1.

Data were collected during COVID-19 pandemic restrictions, where lockdowns and fears prohibited many patients from accessing mental health care. This resulted in a smaller sample size and elongated the data collection period (October 2021 to March 2023). Before data collection, researchers explained the study's objectives, potential risks and benefits, and participants' right to withdraw. All the participants provided written informed consent. The project received ethical approval prior to data collection from the Ethics Committee of the Department of Clinical Psychology, University of Dhaka (protocol# MS210504, approved on 18 May 2021).

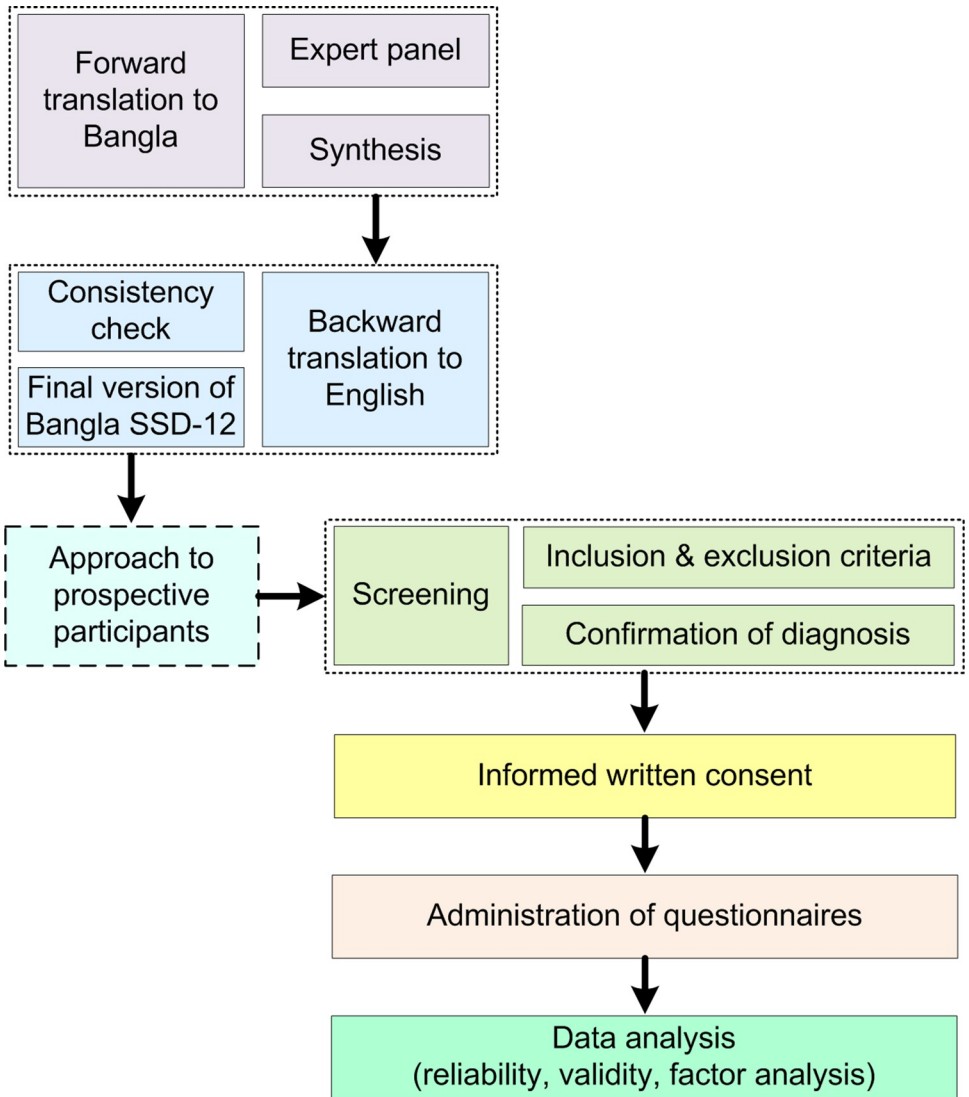

**Fig 1. Flow chart of the process involved in the translation and validation of SSD-12.**

## Findings

### Validity

The content in the items of the Bangla SSD-12 went through rigorous evaluation during forward and backward translation. As no changes other than translation were made to the original English version with known content validity [20], the Bangla SSD-12 can be claimed to have content validity.

The criterion validity of the Bangla SSD-12 was evaluated using its correlation with the Somatic Complain Subscale (SCS) of the Morey Personality Inventory [25, 26]. The SCS was considered as the criterion measure and a high correlation r = .86 (p < .01) was found between scores of the Bengali SSD-12 and the SCS (Table 1).

Diagnostic status was used as an additional criterion for assessing criterion validity of the Bangla SSD-12 so independent samples t-test was used to compare the mean SSD-12 scores of

**Table 1. SSD-12 correlations with SCS, HADS, and its anxiety and depression subscales.**

| Type of Validity | Instruments | Coefficient | Cohen's d |
|---|---|---|---|
| Criterion | Somatic complain scale (SCS) | r = .86** | 3.34 |
| Criterion | Somatization diagnosis | $t_{198} = 16.74$** | 2.37 |
| Construct (Convergent) | Psychological distress (HADS) | r = .67** | 1.79 |
| Construct (Convergent) | Anxiety subscale of HADS | r = .64** | 1.65 |
| Construct (Convergent) | Depression subscale of HADS | r = .57** | 1.37 |

**p < .01.

clinical and non-clinical samples. Significant difference ($t_{198} = 16.74$, p < .01) between scores for clinical (M = 31.61, SD = 7.60) and non-clinical (M = 11.81, SD = 9.06) groups was found. Additionally, to assess the diagnostic accuracy of the SSD-12, the Receiver Operating Characteristic (ROC) curve was used (Fig 2).

The area under the curve was .936 (95% CI [.903, .969], p < .01), indicating excellent diagnostic performance. The sensitivity-specificity calculation suggests the optimal cutoff score of 22 with a sensitivity of 0.89 and a specificity of 0.84 (Table 2).

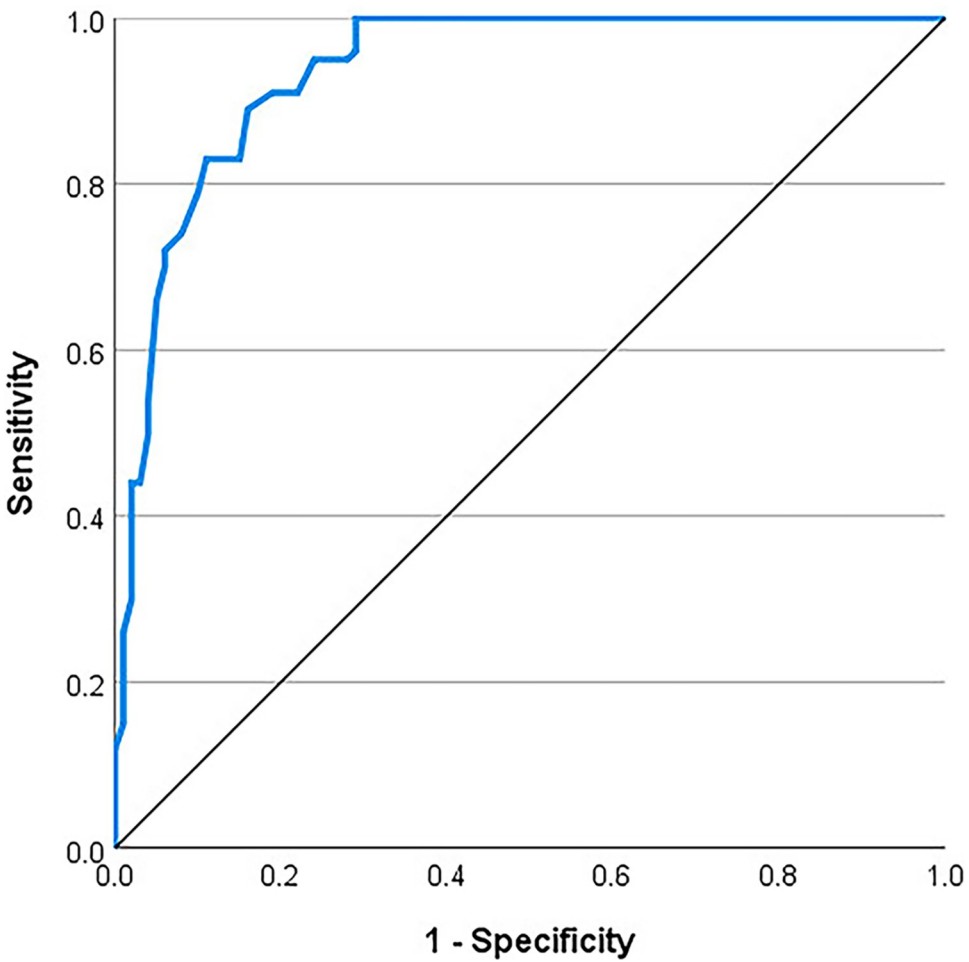

**Fig 2. Receiver Operating Characteristic (ROC) curve for SSD-12.**

**Table 2. Sensitivity and specificity of the SSD-12 in the intermediate range.**

| Score | Sensitivity | Specificity |
|---|---|---|
| 13.50 | 1.00 | .710 |
| . . . | | |
| 19.50 | .910 | .790 |
| 20.50 | .910 | .810 |
| 22.00 * | .890 | .840 |
| 23.50 | .830 | .850 |
| 24.50 | .830 | .890 |
| . . . | | |
| 37.50 | .260 | .010 |

* Optimal cutoff score

The convergent method was used to assess construct validity of the Bangla SSD-12. As somatic problems are known to increase the psychological burden of a person [20], it was hypothesized that people with higher somatic complaints would have higher levels of psychological distress, anxiety and depression compared to those without such complaints [31]. Similar to previous studies, the HADS was used to assess construct validity in this regard [27]. The SSD-12 total score was found to have adequate positive correlations with the overall HADS score (r = .67, p < .01), its anxiety subscale (r = .64, p < .01), and its depression subscale (r = .57, p < .01) (Table 1).

## Reliability

Internal consistency reliability of the SSD-12 was tested through three indicators; corrected item-total correlation, Cronbach's alpha and split-half correlation. The corrected item-total correlation demonstrated adequate correlation of the individual items with the corrected total scores (average r = .731; range r = .463 to .817) (see Table 3). The overall Cronbach's alpha for the 12-item scale was .94, indicating high internal consistency of the translated instrument

**Table 3. Itemized statistics for internal consistency and factor analysis.**

| | Items | Corrected item-total correlation | Factor loading |
|---|---|---|---|
| 1 | I think that my physical symptoms are signs of a serious illness | .728 | .775 |
| 2 | I am very worried about my health | .638 | .695 |
| 3 | My health concerns hinder me in everyday life | .761 | .807 |
| 4 | I am convinced that my symptoms are serious | .790 | .832 |
| 5 | My symptoms scare me | .810 | .849 |
| 6 | My physical complaints occupy me for most of the day | .783 | .824 |
| 7 | Others tell me that my physical problems are not serious | .463 | .517 |
| 8 | I'm worried that my physical complaints will never stop | .788 | .833 |
| 9 | My worries about my health take my energy | .798 | .839 |
| 10 | I think that doctors do not take my physical complaints seriously | .613 | .671 |
| 11 | I am worried that my physical symptoms will continue into the future | .783 | .825 |
| 12 | Due to my physical complaints, I have poor concentration on other things | .817 | .856 |

(Table 3). Additional support for the internal consistency of the instruments came from split-half reliability which was calculated using the odd-even method and demonstrated a strong correlation (Spearman-Brown r = .93).

### Factor analysis

**Confirmatory Factor Analysis (CFA).**   As the original scale has been known to have a three-factor structure [20], we conducted a confirmatory factor analysis to assess the fit of the known factor structure with Bangladeshi data. The three-factor measurement model (see Fig 3) was tested using AMOS 18 software program [32]. Several fit indices were consulted to assess the adequacy of the three-factor model, including Chi-square ($\chi^2$), the ratio of Chi-square to degrees of freedom ($\chi^2$/df), root mean square error of approximation (RMSEA), comparative fit index (CFI), Tucker-Lewis index (TLI), and standardized root mean square

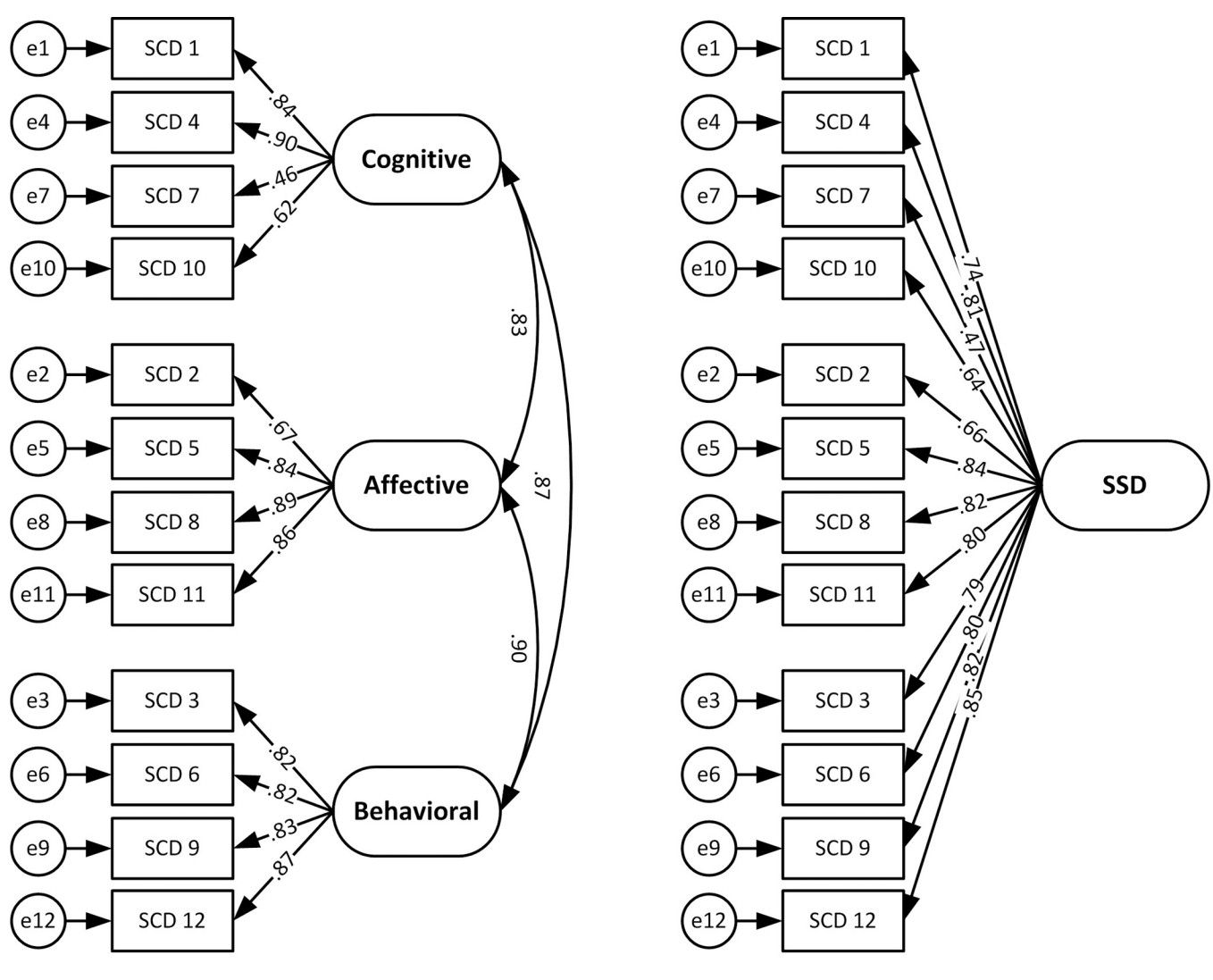

**Fig 3. Fitness of the measurement model showing loadings on two alternative factor structures.**

**Table 4. The goodness of fit indices for three-factor and one-factor model of SSD-12 measurement structure.**

|  | $\chi^2$ | df | p | $\chi^2$/df | RMSEA [CI] | CFI | TLI | SRMR |
|---|---|---|---|---|---|---|---|---|
| Three-factor model | 245.11 | 51 | .01 | 4.81 | .138 [.121, .156] | .895 | .864 | .058 |
| One-factor model | 317.97 | 54 | .01 | 5.89 | .157 [.140, .174] | .857 | .825 | .055 |

residual (SRMR). The criteria for evaluating goodness of fit were as follows: $\chi^2$ with p ≥ .01, $\chi^2$/df ≤ 2, RMSEA ≤ .06, CFI ≥ .95, TLI ≥ .95, and SRMR ≤ .08 [see 33]. None of the indices suggested fit of the three-factor model ($\chi^2$ = 245.11, p < .01; $\chi^2$/df = 4.81; RMSEA = .138; CFI = .895; TLI = .864, and SRMR = .058) for the Bangla SSD-12 (Table 4). Subsequently, an exploratory factor analysis was carried out to identify the factor structure of the Bangla SSD-12.

**Exploratory Factor Analysis (EFA).** The Kaiser-Meyer-Olkin value (.91) and Bartlett's test of sphericity ($\chi^2$ = 1864.32, p < .01) indicated sampling adequacy and suitability of the data for factor analysis [34, 35]. The Eigenvalues and scree plot (Fig 4) suggested one factor, accounting for 61.29% of the total variance. All the items displayed adequate factor loadings for their retention under the single factor (see Table 3).

The single-factor model suggested by exploratory factor analysis was further tested using single-factor measurement model in the AMOS program [32], however, the indices indicated lack of fit (see Fig 3).

## Discussion

The DSM-5 outlines three criteria for diagnosing somatic symptom disorder among which Criterion-B deals with its manifestations through disproportionate thoughts, feelings, and behaviors related to somatic symptoms. Although the Structured Clinical Interview for DSM-5

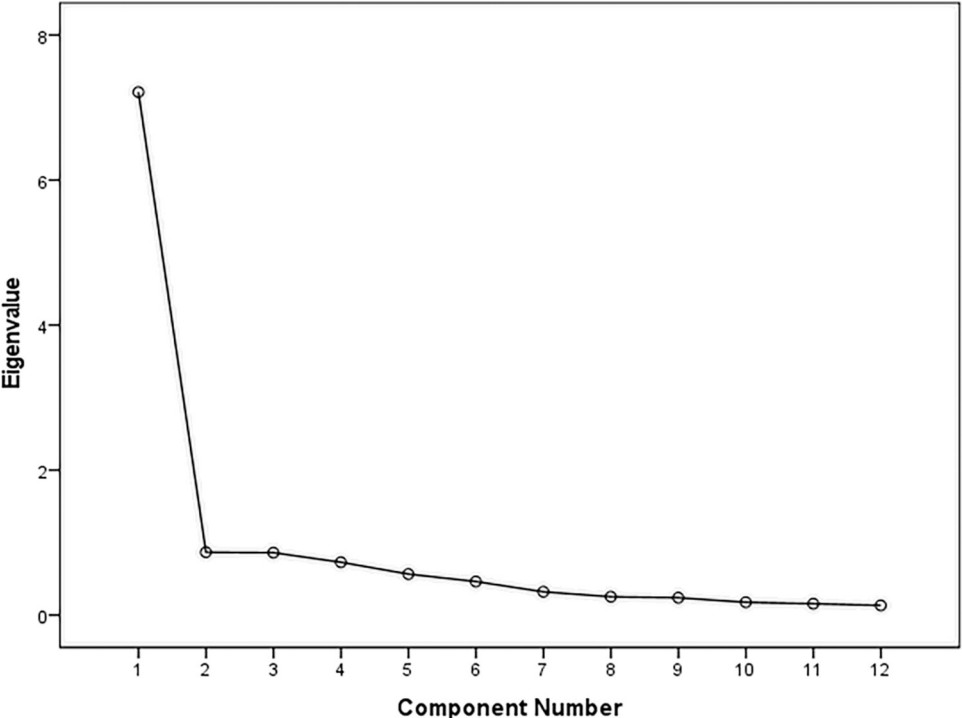

**Fig 4. Scree plot from EFA for SSD-12.**

(SCID-5) [36] is considered the standard method for the diagnosis of somatic symptoms, it is often taxing for clinicians to evaluate the B criteria in daily clinical work due to its time-consuming nature and inflexibility in clinical settings [20]. Diagnosing and treating somatically preoccupied patients is challenging, as they often prioritize physical concerns over psychological and social issues. In addition, it is crucial to address comorbid conditions such as substance abuse, depression, and anxiety. Due to the competing demands of treating and managing multiple conditions in the same patients, primary care practitioners often find it taxing to diagnose and manage patients with somatic symptom disorders [37]. Diagnosing these disorders thus presents significant challenges and requires comprehensive skills and knowledge of the biopsychosocial spectrum [37]. Consequently, patients with somatic symptom disorders are frequently referred from primary care to multiple specialists, creating a challenging cycle that is difficult to break. This inefficient use of resources can hinder appropriate care and exacerbate further patients' symptoms and sufferings [38].

In Bangladesh, no other instruments are available that directly assess the three features of DSM-5 Somatic Symptom Disorder, particularly for criterion B somatic symptoms. Therefore, this study aimed to translate and validate the Bangla version of the SSD-12 to aid in the prompt diagnosis and assessment of therapeutic outcomes.

The findings demonstrated that the Bangla SSD-12 is a reliable and valid instrument for assessing somatic symptom disorder. Correlations between the SSD-12 scores and measures of somatic complaints, anxiety, and depression were moderate to high, affirming the test's criterion and construct validity of the instrument. The findings are aligned with validation studies conducted in other Asian and European countries [20, 39–41]. A unique feature of the present study is that it also demonstrated the SSD-12's ability to discriminate between clinical and non-clinical samples ($t_{198} = 16.74$, p < .01) adding further support for its criterion validity. The ROC analysis indicates excellent diagnostic performance (area under the curve .936) of the SSD-12. With an optimal cutoff score of 22, the scale demonstrates acceptable sensitivity (89%) and specificity (84%). Similar values for cutoff and corresponding sensitivity and specificity of the has been reported in other contexts too [39].

The comorbidity of depression and anxiety with somatization is well-known [29]. Therefore, to measure convergent validity, the HADS with its anxiety and depression subscales was used. In line with previous study findings, [29, 39] the Bangla SSD-12 also demonstrated strong convergent validity through high correlations with the HADS and its subscales.

The Cronbach's alpha (.94) of the Bangla version of the SSD-12 indicates strong internal consistency of the tool. It may be noted here that Cronbach's alpha is highly influenced by the number of items in the measure. Therefore, a high alpha for a moderate-length (12-item) tool is a clear indication of the truly strong internal consistency of the Bangla SSD-12. Additionally, the split-half reliability (spearman r = .95) and the corrected item-total correlation (r = .46 to .82 for individual items) provide further support to its internal consistency reliability. The high internal consistency reliability of the Bangla SSD-12 is consistent with the original instrument [20] and other validation studies [39, 41]. These findings collectively support the robustness and reliability of the Bangla version of the SSD-12 in the Bangladesh context.

While most of the previous studies have supported a three-factor model (i.e., cognitive, affective, and behavioral) for the SSD-12 [20, 39], confirmatory factor analysis with Bangladesh data indicated a lack of fit across all indices using the stringent fit criteria suggested in recent validation studies conducted in the context [33]. Subsequent, exploratory factor analysis, identified a single-factor structure explaining 61.29% of the total variance. The single-factor structure of the tool was also reported in the Dutch context [41]. When the single-factor measurement model was put to the test using the present data, it did not show model fit on the indices. The divergence between EFA and CFA results is not uncommon in psychometric

research [42, 43]. Being exploratory, EFA tends to identify dominant patterns within the data, which may oversimplify complex structures [44]. In contrast, CFA imposes strict constraints, such as requiring all items to load exclusively onto a single factor and setting non-target loadings to zero, making it more sensitive to model misspecifications [43]. These methodological differences may explain why the single-factor model identified in the EFA failed to achieve acceptable fit in the CFA. The differences in the factor structure between Bangladesh and other countries warrant further investigation to understand the universality of item interpretations.

The Bangla SSD-12 exhibits excellent psychometric properties comparable to other instruments available for assessing somatic symptoms, such as the Bangla Morey's Somatic Complaints Scale [25]. Adequate psychometric properties are important features contributing to the confident use of any instrument. Translated instruments with psychometric properties similar to the Bangla SSD-12, such as the Perceived Stress Scale [33], the Bangla Mindful Attention Awareness Scale [45] and the Bangla WHO-5 Well-being Index [46] are being widely used in Bangladesh. Therefore, it is also likely that the Bangla SSD-12 will be accepted among researchers and clinicians in Bangladesh.

One of the constraints in this study was the use of a small sample size. However, a sample size of 200 in this analysis can be statistically justified as it is far above the suggested sample size based on the minimum subject-to-item ratio of 5:1[47]. The clinical sample was drawn from three tertiary referral hospitals situated in the country's capital with the assumption that the sample would reflect participants from across the country. Exploring the SSD-12's utility as a screening tool in primary care settings and its predictive capabilities would be valuable. To establish its screening and severity norm, participants from a wider range of geographic regions within the country can be used in future research. Collecting evidence of validity is an ongoing process for psychological instruments. Further work on the validation of the Bangla SSD-12 will enhance its robustness as well as confidence among its users regarding the utility of the instrument.

## Conclusions

The thorough process followed in translating the SSD-12 in Bangla contributed to the strong psychometric properties that established it as a reliable and valid instrument. It is expected to fill the gap in the assessment and monitoring of somatic symptom disorder in clinical as well as research contexts. The SSD-12 being a fast and easy to administer tool, appeals for its use as a practical tool in hectic clinical environments. The use of this tool can be ultimately linked with improved patient outcomes. Future research should explore the scale's applicability in diverse clinical populations within the region and its potential for tracking and providing feedback related to treatment outcomes.

## Author Contributions

**Conceptualization:** Sumaiya Habib.

**Data curation:** Sumaiya Habib.

**Formal analysis:** Sumaiya Habib, Muhammad Kamruzzaman Mozumder.

**Methodology:** Sumaiya Habib, Muhammad Kamruzzaman Mozumder.

**Supervision:** Muhammad Kamruzzaman Mozumder.

**Visualization:** Muhammad Kamruzzaman Mozumder.

**Writing – original draft:** Sumaiya Habib.

**Writing – review & editing:** Sumaiya Habib, Muhammad Kamruzzaman Mozumder.

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
