## [Decision Letter · Decision Letter 0]

26 Nov 2024

PONE-D-24-42215Validation of the Somatic Symptom Disorder - B Criteria Scale (SSD-12) in BangladeshPLOS ONE

Dear Dr. Mozumder,

Thank you for submitting your manuscript to PLOS ONE. After careful consideration, we feel that it has merit but does not fully meet PLOS ONE’s publication criteria as it currently stands. Therefore, we invite you to submit a revised version of the manuscript that addresses the points raised during the review process.

As a point to note, when revising your paper, please respond to all comments.

We look forward to receiving your revised manuscript.

Kind regards,

Kenta Matsumura

Academic Editor

PLOS ONE

2. Thank you for stating the following financial disclosure: [Collection of data was partially funded by Bangladesh National Science and Technology (NST) Fellowship received by SH. However, the author(s) did not receive any funding for preparing or publishing the manuscript.]. At this time, please address the following queries: a) Please clarify the sources of funding (financial or material support) for your study. List the grants or organizations that supported your study, including funding received from your institution. b) State what role the funders took in the study. If the funders had no role in your study, please state: “The funders had no role in study design, data collection and analysis, decision to publish, or preparation of the manuscript.” c) If any authors received a salary from any of your funders, please state which authors and which funders. d) If you did not receive any funding for this study, please state: “The authors received no specific funding for this work.” Please include your amended statements within your cover letter; we will change the online submission form on your behalf.

Additional Editor Comments:

I understand that it is not the original three-factor structure, but a one-factor structure. However, because the description of this verification process is omitted, I cannot judge whether the results support the conclusion. Therefore, I request the following items be added.

- Presentation of the scree plot

- Report of each fit index when applying CFA to the one-factor structure (please be careful not to set correlation between errors when analyzing)

The authors state that cases can be discriminated based on the results of the t-test, but ROC analysis would be more appropriate here. Therefore, please add the following.

- ROC (figure)

- Typical values such as AUC and its 95% confidence interval, sensitivity, specificity, positive predictive value, etc.

Other

・If possible, please state the effect size

・L154: “α” —> “p”?

Reviewers' comments:

Reviewer's Responses to Questions

**Comments to the Author**

1. Is the manuscript technically sound, and do the data support the conclusions?

Reviewer #1: Yes

2. Has the statistical analysis been performed appropriately and rigorously? 

Reviewer #1: Yes

3. Have the authors made all data underlying the findings in their manuscript fully available?

Reviewer #1: Yes

4. Is the manuscript presented in an intelligible fashion and written in standard English?

Reviewer #1: Yes

5. Review Comments to the Author

Reviewer #1: This study represents an important step toward the diagnosis of Somatic Symptom Disorder. However, I believe some revisions are necessary to enhance its clarity and impact.

General Comments

I recommend providing a more comprehensive description of Somatic Symptom Disorder to contextualize the study. Including the following information will help convey the significance and importance of this research.

Abstract

Please ensure consistency in terminology throughout the paper. For example, on lines 18 and 19, the terms "Non-clinical" and "nonclinical" are used interchangeably. Standardizing the format will improve readability and precision.

Introduction

A more detailed explanation of the three diagnostic criteria for Somatic Symptom Disorder is needed. Additionally, I recommend including the following information:

Typical symptoms

Prevalence rates

Comorbidities with other psychiatric disorders

Specific disease names and characteristics of individuals prone to developing the disorder

Treatment duration, methods, and prognosis

Providing these details will offer readers a clearer and more comprehensive understanding of the disorder.

Methods

Demographic Details: Include information about the average duration of Somatic Symptom Disorder in the clinical group (e.g., mean value, standard deviation). Address potential biases in the patient groups across the three facilities.

Questionnaire Details: Incorporate some of the specific questions and characteristics from the three domains of Somatic Symptom Disorder in the text, not just in the tables, to enhance clarity.

Instructions to Respondents: Provide information on the instructions given to respondents. For instance, clarify whether participants were asked to answer based on their experiences over the past week or the past few days.

Discussion

You mention that diagnosing criterion B is burdensome, but only one study is cited to support this claim. I recommend:

Discussing this study in more detail to provide a stronger foundation for your argument.

Including additional relevant citations to strengthen the discussion.

6. PLOS authors have the option to publish the peer review history of their article (what does this mean?). If published, this will include your full peer review and any attached files.

Reviewer #1: **Yes: **Mariko Inoue

---

## [Author Response · Author response to Decision Letter 0]

8 Jan 2025

Response to Reviewer Comments to the Author

A. Reviewer #1: 

This study represents an important step toward the diagnosis of Somatic Symptom Disorder. However, I believe some revisions are necessary to enhance its clarity and impact.

We sincerely thank you for your thoughtful comments and valuable suggestions to improve our manuscript. We have carefully addressed each of your points and made corresponding revisions to the manuscript as detailed below.

General Comments: I recommend providing a more comprehensive description of Somatic Symptom Disorder to contextualize the study. Including the following information will help convey the significance and importance of this research.

Response: We have revised the introduction section thoroughly as per this and other comments and we believe it now covers a more comprehensive description of Somatic Symptom Disorder to contextualize the study.

Abstract: Please ensure consistency in terminology throughout the paper. For example, in lines 18 and 19, the terms "Non-clinical" and "nonclinical" are used interchangeably. Standardizing the format will improve readability and precision.

Response: Thank you for noticing this inconsistency. We have checked the manuscript and changed all such instances to "non-clinical" for consistency. 

Introduction: A more detailed explanation of the three diagnostic criteria for Somatic Symptom Disorder is needed. Additionally, I recommend including the following information:

 Typical symptoms

 Prevalence rates

 Comorbidities with other psychiatric disorders

 Specific disease names and characteristics of individuals prone to developing the disorder

 Treatment duration, methods, and prognosis

Providing these details will offer readers a clearer and more comprehensive understanding of the disorder.

Response: We appreciate this suggestion and have expanded the Introduction section to include a more detailed description of Somatic Symptom Disorder (SSD). The following details have been added to the introduction section.

Somatization is generally characterized by three distinct features [6]. Firstly, it presents with medically unexplained somatic symptoms such as pain, weakness, or shortness of breath, Secondly, patients exhibit somatic preoccupation or hypochondriacal worry, demonstrating excessive engagement (in terms of time and effort) and concerns about their bodily symptoms. These concerns are often overtly disproportionate and may be accompanied by excessive body checking and reassurance-seeking [7]. Thirdly, patients may present these somatic symptoms as part of the clinical manifestations of anxiety, affective, or other psychiatric disorders [6]. (see page 3)

Somatization may present with a range of symptoms, Bangladeshi children and adolescents diagnosed with somatoform disorder report an average of 12 to 16 somatic symptoms [15]. Pain is the most common symptom among patients. In Bangladesh, abdominal pain is the most frequent somatic symptom in children and adolescents, while a German study of 7,925 adults aged 40 to 80 years found that pain complaints (arms, legs, joints, back pain) were most common, followed by back pain, headaches, nausea, constipation/diarrhea, shortness of breath, dizziness, and heart racing or pounding. (see page 4)

Across the globe, including Bangladesh, somatic symptom disorders have been reported to be the third most prevalent mental disorders followed by depression and anxiety [10]. (see page 4)

 Somatoform disorders are among the most common psychiatric disorders in general medical settings. Somatoform disorders often coexist with other comorbidities, with 8% of primary care patients meeting the criteria for 'multi-somatoform disorder’[13]. Depression, conversion disorder, hypochondriasis, somatization, and pain disorders are the most common comorbid conditions associated with somatization[14]. Apart from comorbidity, somatization disorder has been reported to be associated strongly with depression and anxiety, moderately with schizophrenia and mania, and weakly with substance use and antisocial personality[15].(see page 4)

Accurate diagnosis, support, and reassurance are the cornerstones of the treatment of somatization disorder [16]. Approaches typically involve psychotropic medications and psychological therapies (such as cognitive behavior therapy) focusing on cognitive, emotional, and behavioral aspects [17]. The prognosis for somatic symptom disorder shows improvement in 50–75% of patients, while 10–30% experience a worsening of their condition under combination treatment (medication and psychological therapy)[18]. Fewer symptoms and higher baseline functioning have been linked with a better prognosis [19]. (see page 4-5)

Methods

Demographic Details: Include information about the average duration of Somatic Symptom Disorder in the clinical group (e.g., mean value, standard deviation). Address potential biases in the patient groups across the three facilities.

Response: Unfortunately, we did not record the average duration of symptoms during data collection and hence are unable to present the suggested data. Thank you for your valuable comment it will help us improve data collection in future research. To address the bias, we have added the following details in the participant section.

Data for this study were collected from the psychiatric departments of three tertiary hospitals, providing access to a diverse patient population. For one of the hospitals, the majority of the data were collected from the inpatient department, while for another, the majority came from the outpatient department. These differences may result in inadvertent variation in the patient groups across the three settings. However, as no comparisons between the settings were planned, we believe these variations contribute to the generalizability of the findings. (see page 7)

Questionnaire Details: Incorporate some of the specific questions and characteristics from the three domains of Somatic Symptom Disorder in the text, not just in the tables, to enhance clarity.

Response: We agree with this suggestion and have included sample questions from the three domains (somatic symptoms, excessive thoughts, and behaviors) directly in the Instruments section to enhance clarity.

The respondents report their experiences of cognition, emotion, or behavior on a 5-point Likert scale such as, “I think that my physical symptoms are signs of a serious illness”, “I am very worried about my health”, “My health concerns hinder me in everyday life”. (see page 7)

Instructions to Respondents: Provide information on the instructions given to respondents. For instance, clarify whether participants were asked to answer based on their experiences over the past week or the past few days.

Response: We have clarified the instructions provided to participants in the Instruments section.

Participants were instructed to complete the SSD-12 questionnaire based on their experiences over the past week, aiming to capture the severity of recent symptoms and their impact. (see page 7)

Discussion: You mention that diagnosing criterion B is burdensome, but only one study is cited to support this claim. I recommend: Discussing this study in more detail to provide a stronger foundation for your argument. Including additional relevant citations to strengthen the discussion.

Response: We have expanded the discussion of the cited study to better support the claim regarding the burden of diagnosing criterion B. Additionally, we have included two more relevant studies to strengthen this point.

Diagnosing and treating somatically preoccupied patients is challenging, as they often prioritize physical concerns over psychological and social issues. In addition, it is crucial to address comorbid conditions such as, substance abuse, depression, and anxiety. Due to the competing demands of treating and managing multiple conditions in the same patients, primary care practitioners often find it taxing to diagnose and manage patients with somatic symptom disorders [37]. Diagnosing these disorders thus presents significant challenges and requires comprehensive skills and knowledge of the biopsychosocial spectrum [37]. Consequently, patients with somatic symptom disorders are frequently referred from primary care to multiple specialists, creating a challenging cycle that is difficult to break. This inefficient use of resources can hinder appropriate care and exacerbate further patients' symptoms and sufferings [38]. (see page 15-16)

B. Editor Comments

Response: We have checked manuscript again for PLOS One’s style requirements. 

2. Thank you for stating the following financial disclosure: [Collection of data was partially funded by Bangladesh National Science and Technology (NST) Fellowship received by SH. However, the author(s) did not receive any funding for preparing or publishing the manuscript.]. At this time, please address the following queries: a) Please clarify the sources of funding (financial or material support) for your study. List the grants or organizations that supported your study, including funding received from your institution. b) State what role the funders took in the study. If the funders had no role in your study, please state: “The funders had no role in study design, data collection and analysis, decision to publish, or preparation of the manuscript.” c) If any authors received a salary from any of your funders, please state which authors and which funders. d) If you did not receive any funding for this study, please state: “The authors received no specific funding for this work.” Please include your amended statements within your cover letter; we will change the online submission form on your behalf.

Response: Thank you for providing detailed instructions regarding the financial disclosure statement. We have amended it accordingly and added the following text in the cover letter.

Our study was partially funded by the National Science and Technology (NST) Fellowship, from the Ministry of Science and Technology, Government of Bangladesh awarded to SH. The funders had no role in study design, data collection and analysis, decision to publish, or preparation of the manuscript. The authors received no specific funding for this work.

Response: We have updated the Methods section of the manuscript to include the full ethics statement, as follows:

All the participants provided written informed consent. The project received ethical approval prior to data collection from the Ethics Committee of the Department of Clinical Psychology, University of Dhaka (protocol# MS210504, approved on 18 May 2021). (see page 9)

Additional Editor Comments: I understand that it is not the original three-factor structure, but a one-factor structure. However, because the description of this verification process is omitted, I cannot judge whether the results support the conclusion. Therefore, I request the following items be added.

- Presentation of the scree plot 

 Response: Added as Fig 4. (see page 15)

- Report of each fit index when applying CFA to the one-factor structure (please be careful not to set correlation between errors when analyzing) 

Response: We have added a description of the verification process for the one-factor structure, including a scree plot and fit indices (CFI, TLI, RMSEA, SRMR) obtained from the CFA analysis. Correlations between errors were not set, as per the suggestion.

A new figure (Fig 3) depicting the single-factor as well as the three-factor model has been added along with a new table (Table 4) to accommodate the suggestion. Additionally the following texts has been added in result section (see page 15).

The single-factor model suggested by exploratory factor analysis was further tested using single-factor measurement model in the AMOS program [32], however, the indices indicated lack of fit (see Fig 3). (see page 15)

The discussion section has also been updated with the following texts (see page -18).

When the single-factor measurement model was put to the test using the present data, it did not show model fit on the indices. The divergence between EFA and CFA results is not uncommon in psychometric research [42, 43]. Being exploratory, EFA tends to identify dominant patterns within the data, which may oversimplify complex structures [44]. In contrast, CFA imposes strict constraints, such as requiring all items to load exclusively onto a single factor and setting non-target loadings to zero, making it more sensitive to model misspecifications [43]. These methodological differences may explain why the single-factor model identified in the EFA failed to achieve acceptable fit in the CFA. (see page 18) 

The authors state that cases can be discriminated based on the results of the t-test, but ROC analysis would be more appropriate here. 

Response: As per suggestion we have added ROC analysis and relevant texts has been added as follows. 

In the result section (see page 10 )

Additionally, to assess the diagnostic accuracy of the SSD-12, the Receiver Operating Characteristic (ROC) curve was used (Fig 2).

The area under the curve was .936 (95% CI [.903, .969], p < .01), indicating excellent diagnostic performance. The sensitivity-specificity calculation suggests the optimal cutoff score of 22 with a sensitivity of 0.89 and a specificity of 0.84 (Table 2). (see page 10)

And in the discussion section (see page 17)

The ROC analysis indicates excellent diagnostic performance (area under the curve .936) of the SSD-12. With an optimal cutoff score of 22, the scale demonstrates acceptable sensitivity (89%) and specificity (84%). Similar values for cutoff and corresponding sensitivity and specificity of the has been reported in other contexts too [39]. (see page 17)

Therefore, please add the following.

- ROC (figure)

Response: ROC (figure) has been added as per suggestion (Fig 2.; page 10).

- Typical values such as AUC and its 95% confidence interval, sensitivity, specificity, positive predictive value, etc.

Response: These have been added as per suggestions please see response to earlier comment). We have also added a table (Table 2) to present the sensitivity and specificity. (see page 11)

Other

・If possible, please state the effect size

Response: We have added effect size estimates (Cohen’s d) as per suggestions in Table 1 (see page 10)

・L154: “α” —> “p”?

Response: We sincerely apologize for the typos, and these have been corrected as per suggestion. Subsequently, we ran checks throughout the whole manuscript again for errors in texts and numbers and realized that we did not update few data values while preparing the draft. We have corrected those inadvertent mistakes too.

---

## [Editor Report · Decision Letter 1]

10 Jan 2025

Validation of the Somatic Symptom Disorder - B Criteria Scale (SSD-12) in Bangladesh

PONE-D-24-42215R1

Dear Dr. Mozumder,

We’re pleased to inform you that your manuscript has been judged scientifically suitable for publication and will be formally accepted for publication once it meets all outstanding technical requirements.

Kind regards,

Kenta Matsumura

Academic Editor

PLOS ONE
---

## [Editor Report · Acceptance letter]

14 Jan 2025

PONE-D-24-42215R1 

PLOS ONE

Dear Dr. Mozumder, 

I'm pleased to inform you that your manuscript has been deemed suitable for publication in PLOS ONE. Congratulations! Your manuscript is now being handed over to our production team.

Kind regards, 

on behalf of

Dr. Kenta Matsumura 

Academic Editor

PLOS ONE
